# Accelerating Recovery: A Case Report on Telerehabilitation for a Triathlete’s Post-Meniscus Surgery Comeback

**DOI:** 10.3390/healthcare13040406

**Published:** 2025-02-13

**Authors:** Olimpio Galasso, Mariaconsiglia Calabrese, Giuseppe Scanniello, Marina Garofano, Lucia Pepe, Luana Budaci, Gaetano Ungaro, Gianluca Fimiani, Placido Bramanti, Luigi Schiavo, Francesco Corallo, Maria Pagano, Irene Cappadona, Alessandro Crinisio, Alessia Bramanti

**Affiliations:** 1Department of Medicine, Surgery and Dentistry, University of Salerno, Via S. Allende, 43, 84081 Baronissi, Italy; ogalasso@unisa.it (O.G.); macalabrese@unisa.it (M.C.); abramanti@unisa.it (A.B.); 2Department of Computer Science, University of Salerno, Via Giovanni Paolo II, 132, 84084 Fisciano, Italy; gscanniello@unisa.it (G.S.); gfimiani@unisa.it (G.F.); lschiavo@unisa.it (L.S.); 3Azienda Ospedaliero-Universitaria “San Giovanni di Dio e Ruggi d’Aragona”, Via San Leonardo, 84125 Salerno, Italy; luciapepe88@gmail.com (L.P.); lu.budaci@gmail.com (L.B.); gaungaro@unisa.it (G.U.); 4Faculty of Psychology, University eCampus, 22060 Novedrate, Italy; bramanti.dino@gmail.com; 5Istituto di Ricovero e Cura a Carattere Scientifico Centro Neurolesi Bonino-Pulejo, Via Palermo, S.S. 113, C.da Casazza, 98124 Messina, Italy; francesco.corallo@irccsme.it (F.C.); maria.pagano@irccsme.it (M.P.); irene.cappadona@irccsme.it (I.C.); 6Orthopedic Department “Clinica Ortopedica”, San Giovanni di Dio and Ruggi d’Aragona University Hospital, 84131 Salerno, Italy; alessandrocrinisio@gmail.com

**Keywords:** telerehabilitation, meniscus surgery, virtual reality rehabilitation, post-surgical recovery, athletic rehabilitation, wearable devices

## Abstract

**Introduction:** Meniscus injuries are common among endurance athletes, requiring structured rehabilitation to restore function and facilitate a safe return to sport. Traditional in-person rehabilitation may not always be accessible or feasible for high-performance athletes. Telerehabilitation, incorporating virtual reality, motion tracking, and telemonitoring, offers an innovative approach to guided recovery. However, evidence supporting its effectiveness in elite athletes remains limited. **Case presentation:** This case report explores the application of an innovative telerehabilitation program for a 49-year-old triathlete recovering from partial meniscectomy following a medial meniscus tear. The program was structured into three progressive phases over 12 weeks, focusing on restoring range of motion (ROM), muscle strength, and functional stability while gradually reintroducing sports-specific activities. **Results:** By the end of the rehabilitation, the patient achieved full ROM and muscle strength (scoring 5/5 on the Medical Research Council scale for the vastus medialis), along with a pain-free state in both static and dynamic conditions. The integration of telemonitoring devices facilitated detailed monitoring and feedback, enabling personalized adjustments to the rehabilitation protocol. Key milestones included a return to swimming and cycling in Phase 2, reintroduction of running in Phase 3, and a full resumption of triathlon training by week 12. **Conclusions:** Despite the positive results, the study highlights the need for further research to validate these findings across larger cohorts and establish standardized telerehabilitation protocols for athletes. This case underscores the potential of digital health technologies in enhancing recovery trajectories for high-demand athletes post-meniscus surgery, paving the way for supervised, accelerated, and effective sports reintegration.

## 1. Introduction

Although postoperative rehabilitation is crucial for a successful return to sports, there is no consensus regarding postoperative rehabilitation protocol after meniscal surgery. However, the postoperative phase is generally characterized by a variable period of joint immobilization, and restriction of range of motion (ROM) and weight bearing [1,2,3]. In more restricted rehabilitation protocols return to sports is not generally allowed until a minimum of six months post-surgery, while in accelerated protocols patients are often released back to unrestricted sports participation approximately 3–4 months postoperatively [2,4]. An accelerated rehabilitation protocol may be safely implemented for appropriate patients, but further studies are needed to determine an optimal rehabilitation protocol [5].

In this case report, the athlete underwent a tailored rehabilitation program delivered through the Khymeia (Padova, Italy) telerehabilitation system. Telerehabilitation is an innovative approach that utilizes digital communication technologies to deliver rehabilitation services remotely. It gained significant attention during the COVID-19 era, and can also be used to support the return to competition quickly and safely [6,7].

To date, the authors are unaware of any studies that effectively and safely address the return to triathlon after meniscus surgery, particularly following an early telerehabilitation program integrated with a gradual resumption of athletic activity. The individual undergoing rehabilitation in this study was a triathlete who sustained a meniscal injury during training. The lower limbs, which are particularly susceptible to damage in triathlon, are the most affected [8]. Although some literature exists on post-meniscectomy rehabilitation, there is a lack of studies specifically addressing the integration of an early telerehabilitation program for high-performance athletes like triathletes. The provision of a detailed description of this rehabilitation approach therefore contributes to expanding the current knowledge in this field.

### Objective

The primary objective was to demonstrate the feasibility and effectiveness of an early-initiated telerehabilitation program tailored to support a rapid return to sports for a 49-year-old triathlete following meniscus surgery. The case study aimed to highlight the program’s capacity to restore functional performance and expedite recovery in high-demand athletes.

## 2. Case Presentation

### 2.1. Patient Information

The patient was a 49-year-old male triathlete, employed as an associate professor of computer science. He had an extensive background in endurance sports, regularly engaging in triathlon training involving swimming, cycling, and running. In May 2024, the patient sustained a medial meniscus tear during an off-road running session, resulting in swelling and posteromedial knee pain that worsened with activity.

The knee examination showed medial joint line tenderness along with a positive McMurray’s test for medial meniscus. A ten-degree lack of full active and passive flexion was recorded. Plain radiographs of the knee were unremarkable and an MRI study showed an oblique tear through the posterior horn of the medial meniscus. Informed consent was obtained for an arthroscopic procedure. The patient was operated in the supine decubitus position with a pneumatic tourniquet at the root of the thigh (pressure at 250–300 mmHg) to optimize intra-articular visualization. Disinfection was obtained with povidone iodine and sterile drapes were used. Standard anteromedial and anterolateral arthroscopic portals were performed and an injury of the posterior horn of the medial meniscus was noted. Arthroscopy also revealed that there were superficial grade I International Cartilage Repair Society (ICRS) [9] erosions in the central portions of the medial tibial and femoral articular surfaces. No injuries of the anterior and posterior cruciate ligaments were reported. The tear of the medial meniscus appeared irreparable and it was in the inner avascular two-thirds of the meniscus; thus, a selective partial meniscectomy was performed using basket forceps. A probe was used to evaluate mobility and instability of the injured tissue; the meniscal margins were trimmed with a motorized shaver to avoid unstable flaps and guarantee a regular edge.

Weight bearing was allowed as tolerated with the use of crutches until normal gait was achieved two weeks postoperatively [10,11]. Cryotherapy was prescribed for 15 min four times a day and deep vein thrombosis prophylaxis was performed using low-molecular-weight heparin [12,13]. The patient recovered uneventfully.

### 2.2. Preoperative Status

Prior to surgery, the patient maintained a consistent and structured training routine, emphasizing endurance and strength. His regimen typically included multiple weekly sessions spanning the three disciplines of triathlon: swimming (2–3 times a week), cycling (3 times a week), and running (3–4 times a week), with a focus on enhancing cardiovascular fitness and muscular endurance. The described training regimen reflects the athlete’s habitual practice and personal preferences rather than a standardized protocol, it aligns with the typical training loads observed in competitive triathletes but was individually structured by the patient based on his experience and competition goals. Our patient, in fact, was a regular competitor in both national and international races, frequently achieving top rankings within his category, so his primary outcome was to return to training and competition as soon as possible.

### 2.3. Intervention

The technology used for delivering telerehabilitation programs is the Khymeia system (Virtual Reality Rehabilitation System, VRRS of the Khymeia group, Noventa Padovana, Italy; https://khymeia.com/it/; accessed on 9 February 2025), a Class I certified medical device designed for remote rehabilitation, that has been validated in previous studies for its reliability in motor rehabilitation and its effectiveness in telemonitoring-based physiotherapy [14,15,16]. It offers advanced solutions for both patients and clinicians, especially virtual reality environments and motion tracking, which facilitate precise feedback and progress tracking. After an initial in-person training, the telerehabilitation sessions were delivered synchronously, with a frequency of 5 times per week, each lasting one hour, for a total number of 25 sessions. From a nutritional standpoint, the patient adhered to a well-balanced diet during the rehabilitation phase, ensuring adequate intake of both macro- and micronutrients. Proper nutrition throughout this period plays a critical role in enhancing lean body mass, muscular strength, and overall functionality, thereby facilitating a quicker return to daily activities [17,18,19]. In agreement with Smith-Ryan et al. [20], consuming sufficient nutrients prior to rehabilitation sessions optimizes energy availability and exercise performance, while post-session intake supports recovery and physiological adaptation. Consistent with findings from other studies, protein intake was approximately 1.6 g/kg/day [20,21,22]. In particular, starting 3–4 h before a therapy session, the patient was consuming a small meal containing complex carbohydrates (50–100 g) and quality protein (30–40 g) [22]. Beyond protein intake, the diet was supplemented with creatine monohydrate, β-hydroxy-β-methylbutyrate (HMB), omega-3 fatty acids, probiotics, and vitamins to further support muscle mass, strength, and functional improvements during rehabilitation [20,22,23].

## 3. Telerehabilitation Program Details

### 3.1. Phase 1 (Weeks 1–2)

Early mobilization focused on reducing postoperative swelling and regaining range of motion through active exercises (Figure 1). Also, isometric strengthening of the quadriceps and hamstrings was introduced to prevent muscle atrophy. All exercises were performed with weight-bearing and active ROM as tolerated by the patient [10,11,24]. The exercises were performed under real-time monitoring by a physiotherapist connected remotely from the hospital (Figure 2). This was made possible by the patient wearing motion-detection devices, allowing the therapist to assess movement accuracy and provide immediate feedback, ensuring safe and effective execution while adhering to the prescribed limitations. The isometric exercises were also performed with the physiotherapist connected through the Khymeia system; however, these exercises were not included in the system’s database.

#### Physiotherapy Assessment

At the end of phase 1 of the rehabilitation program, the patient showed a 5-degree knee extension deficit, with 90 degrees of flexion. Pain was rated 0 on the VAS scale at rest, while it reached a level of 2 during walking.

### 3.2. Phase 2 (Weeks 3–6)

Phase 2 introduced progressive weight-bearing exercises like half squats and lateral lunges aimed at strengthening the lower extremity muscles (Figure 3). Proprioception training, including the use of balance boards and resistance bands, was added to improve knee stability. Resistance bands were also incorporated into the previously performed exercises to further enhance muscle strength and stability. All exercises performed in this phase are included in the Khymeia system’s database.

At the beginning of the third week and continuing through the fourth, the patient incorporated swimming into his routine, with three training sessions per week, starting with a duration of 10 min and gradually increasing to 30 min. Fins were introduced during swimming sessions to specifically strengthen the vastus medialis. The swimming sessions were monitored using a wearable device, and the reports were sent to the physiotherapist for evaluation of the activities performed. During the fifth and sixth weeks, in addition to the three 30 min swimming sessions, cycling was reintroduced with one 30 min session per week. Even during the cycling sessions, the patient was monitored using a wearable device, with activity reports transmitted to the physiotherapist for evaluation.

#### Physiotherapy Assessment

At the end of Phase 2, the patient achieved full ROM in knee extension and 110 degrees of knee flexion. A strength deficit persisted, particularly in the vastus medialis, which was assessed as 4- on the Medical Research Council (MRC) scale. The patient did not report any pain during movement and no adverse events occurred during the physiotherapy and the training.

### 3.3. Phase 3 (Weeks 7–12)

Dynamic strength training incorporated both closed-chain exercises and plyometric movements to enhance knee stability and strength, which are essential for returning to sport-specific activities. These exercises were not included in the Khymeia system’s database but were taught in person by the physiotherapist and monitored remotely using the system. The patient’s progress was closely monitored on a weekly basis using telerehabilitation tools, allowing the therapist to tailor exercise intensity as needed. Additionally, the patient reintroduced running, starting with one weekly session of 15 min, which included both a warm-up and cool-down to promote gradual adaptation and minimize the risk of re-injury. However, during the running training session in the eighth week, the patient performed a slight rotational movement of the knee near the end of the workout, causing swelling and pain rated at 7 on the VAS scale. Nevertheless, after five days of rest with ice application three times a day, he was able to resume his reconditioning program. After this event, the patient resumed training with cycling and swimming, while running was reintroduced at the beginning of the tenth week. Initially, this included two weekly sessions of approximately 30 min each, during which the patient reported mild discomfort only during the warm-up phase. By the end of the twelfth week, he was able to return to trail running, completing a 12.42 km route (Figure 4).

#### 3.3.1. Physiotherapy Assessment

At the end of Phase 3, the patient achieved full ROM in knee extension and flexion. Muscle strength of the vastus medialis was fully restored, scoring 5/5 on the MRC scale. Pain was rated 0 on the VAS scale in both static and dynamic conditions. The patient successfully resumed all triathlon training sessions (Figure 5), with only a mild imbalance between the left and right side remaining, as evidenced by a ground contact time distribution of 51.4% on the left and 48.6% on the right during a trail running assessment.

#### 3.3.2. Outcomes

The patient’s recovery progress was assessed systematically through ROM, using a goniometer, pain intensity using the VAS [25,26], muscle strength using MRC scale [27] and ground contact time distribution [28,29]. The following Table 1 summarizes these parameters across the preoperative phase and the three rehabilitation phases.

## 4. Discussion

### 4.1. Interpretation of Results

This case report highlights the feasibility and effectiveness of an early-initiated telerehabilitation program tailored to support the recovery of a 49-year-old triathlete following meniscus surgery. The outcomes demonstrate that telerehabilitation, utilizing the Khymeia system, facilitated a safe and progressive return to high-demand athletic activities, such as triathlon training. The patient achieved significant milestones throughout the rehabilitation process:Full ROM in knee extension and flexion by the end of the third phase.Restoration of muscle strength, with the vastus medialis scoring 5/5 on the MRC scale.Pain-free static and dynamic conditions, with no adverse events reported.Successful resumption of triathlon-specific training sessions, including swimming, cycling, and running.

### 4.2. Role of Data Transmission with Wearable Devices

The integration of wearable devices provided valuable real-time data that were instrumental in monitoring and assessing the patient’s recovery progress. These devices [30] are widely accessible and known for their reliability in tracking physical activity [31], captured key metrics such as heart rate [32,33], swimming stroke efficiency [34], cycling cadence [35], training duration, and running biomechanics. This data enabled a detailed analysis of the patient’s physiological responses to rehabilitation exercises and sports-specific activities.

For instance, during swimming sessions, stroke efficiency and duration were evaluated to ensure gradual progression and avoid overexertion. Similarly, cycling cadence and heart rate were monitored to assess cardiovascular performance and ensure that the patient remained within safe exertion levels, finally biomechanical data, like ground contact time, were evaluated during running to identify and address asymmetries, which helped in tailoring exercises to restore balance between the left and right limbs. These metrics were transmitted directly to the physiotherapist, who used the data to tailor the rehabilitation protocol by adjusting exercise intensity, duration, and frequency based on the patient’s recovery status.

By incorporating objective data from the wearable devices, the telerehabilitation program enhanced the precision of exercise prescriptions and provided the patient with clear feedback on their progress. This data-driven approach not only improved adherence to the rehabilitation protocol but also empowered the patient to actively engage in their recovery process. The seamless integration of these devices underscored their utility in facilitating a personalized and adaptive rehabilitation plan, particularly for high-demand athletes [36,37].

The choice of measurement tools can significantly influence the accuracy and reliability of rehabilitation monitoring. Different wearable devices offer varying levels of precision in tracking parameters such as range of motion, muscle activation, and biomechanical data [30]. For instance, some systems provide detailed kinematic analysis, while others focus primarily on step count or heart rate monitoring [36,37]. These variations may impact data interpretation and, consequently, the ability to tailor rehabilitation protocols effectively. Studies have shown that the validity and reliability of wearable devices can differ depending on the activity being measured and the specific technology used [31,37]. Future research should explore how different device functionalities affect rehabilitation outcomes and whether integrating multiple measurement tools could enhance monitoring accuracy.

### 4.3. Comparison with Traditional In-Person Rehabilitation

The outcomes of the telerehabilitation program were compared to benchmarks reported in the literature regarding traditional in-person rehabilitation protocols. Specifically, key metrics such as range of motion (ROM), pain reduction (assessed via VAS), and functional strength (evaluated using the MRC scale) were used to assess the patient’s progress [25,26,27]. These data were contextualized against findings from studies cited in this manuscript to highlight the effectiveness of telerehabilitation relative to traditional approaches.

It should be noted that the patient was not enrolled in a traditional rehabilitation program during this study. Therefore, the comparison presented is based on outcomes reported in the literature rather than a direct side-by-side evaluation of the two approaches. This distinction is crucial for interpreting the results and understanding the broader applicability of telerehabilitation programs [2,4].

## 5. Conclusions and Limitations

This case report illustrates the application of a telerehabilitation program enhanced via the integration of wearable device data to achieve an effective recovery in a triathlete following meniscal surgery. The results highlight the potential for digital health technologies to enhance post-surgical rehabilitation, particularly in high-demand athletic populations, which are not often addressed in standard protocols.

The patient achieved significant milestones, including full ROM, restoration of muscle strength, and a pain-free state, culminating in the successful resumption of triathlon-specific training sessions. The integration of wearable devices provided valuable real-time data, enabling personalized adjustments to the rehabilitation protocol and enhancing the precision of exercise prescriptions.

However, it is important to note that this study is limited by its single-subject design. The findings, while promising, cannot be generalized without further validation across larger cohorts. Future research, including randomized controlled trials, is necessary to establish standardized telerehabilitation protocols for athletes recovering from meniscal surgery. Additionally, the multifactorial nature of recovery, including factors such as diet, exercise adherence, and preoperative physical condition, should be considered in future studies to provide a more comprehensive understanding of the rehabilitation process.

## Figures and Tables

**Figure 1 healthcare-13-00406-f001:**
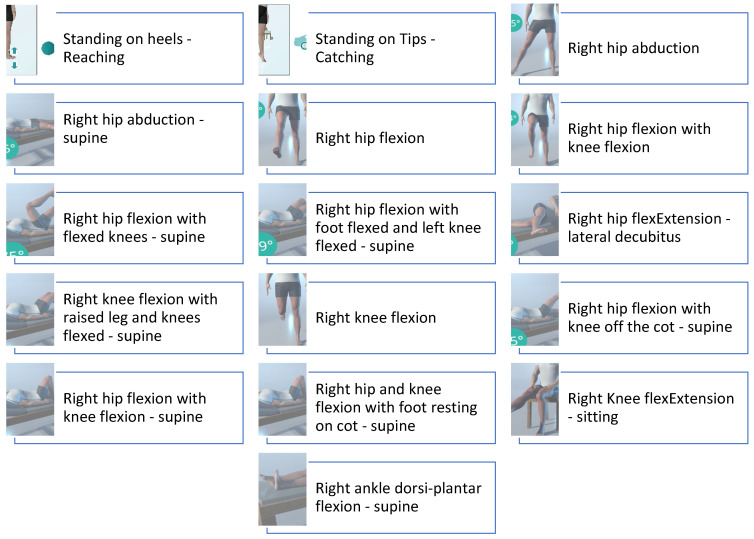
Exercises proposed during Phase 1 and selected from Khymeia database.

**Figure 2 healthcare-13-00406-f002:**
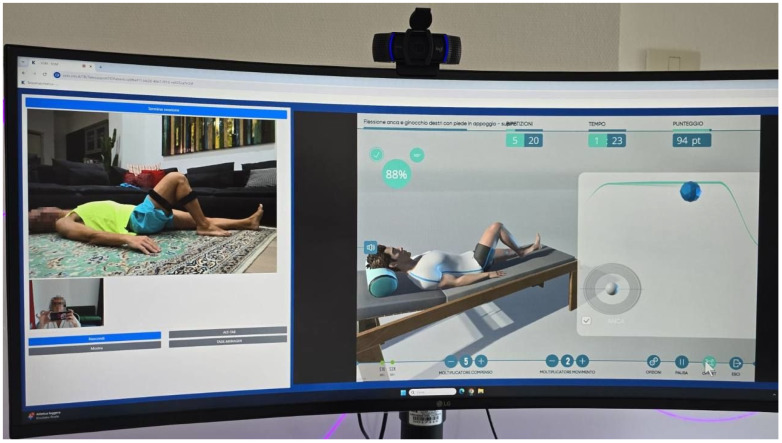
Real-time monitoring and virtual feedback ensure accurate exercise execution for effective remote therapy.

**Figure 3 healthcare-13-00406-f003:**
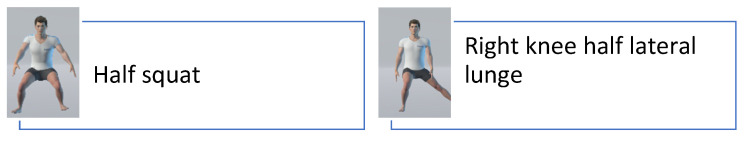
Exercises proposed during Phase 2 and selected from Khymeia database.

**Figure 4 healthcare-13-00406-f004:**
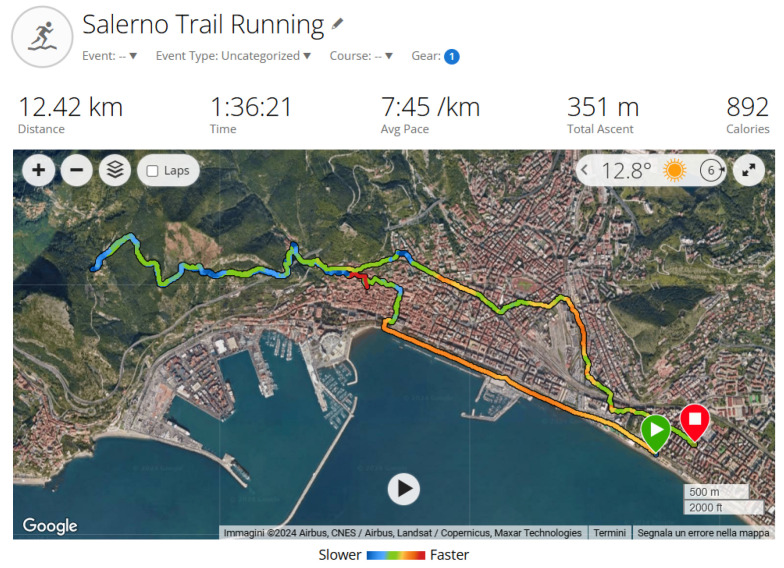
Salerno trail running.

**Figure 5 healthcare-13-00406-f005:**
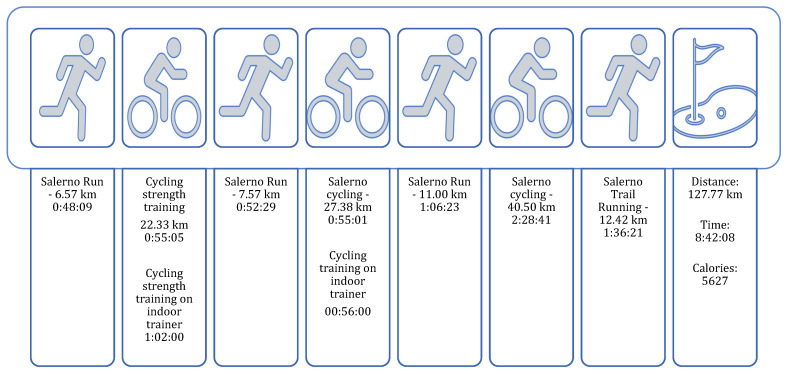
During the twelfth week, the patient resumed all triathlon training sessions.

**Table 1 healthcare-13-00406-t001:** Progression of ROM, Pain Intensity, and Muscle Strength During Rehabilitation.

Phase	Time Period	ROM	Pain Intensity (VAS)	Muscle Strength Left (MRC Scale)	Muscle Strength Right (MRC Scale)	Biomechanical Data (Ground Contact Time Distribution)
Preoperative	Before surgery	Knee flexion: 110°, Knee extension: −10°	3 (walking), 0 (at rest)	Hip flexors: 5/5 Hip extensors: 5/5 Hip abductors: 5/5 Hip adductors: 5/5 Hip external rotators: 5/5 Hip internal rotators: 5/5 Knee extensors: (vastus medialis: 5) Knee flexors: 5/5 Ankle dorsiflexion: 5/5 Ankle plantarflexion: 5/5	Hip flexors: 5/5 Hip extensors: 5/5 Hip abductors: 5/5 Hip adductors: 5/5 Hip external rotators: 5/5 Hip internal rotators: 5/5 Knee extensors: (vastus medialis: 4) Knee flexors: 5/5 Ankle dorsiflexion:/5 Ankle plantarflexion:/5	51.5% left/48.5% right
Phase 1	Weeks 1–2	Knee flexion: 90°, Knee extension: −5°	2 (walking), 0 (at rest)	Hip flexors: 5/5 Hip extensors: 5/5 Hip abductors: 4/5 Hip adductors: 5/5 Hip external rotators: 5/5 Hip internal rotators:/5 Knee extensors: 5 (vastus medialis: 5) Knee flexors: 5/5 Ankle dorsiflexion: 5/5 Ankle plantarflexion: 5/5	Hip flexors: 5/5 Hip extensors: 5/5 Hip abductors: 4/5 Hip adductors: 5/5 Hip external rotators: 5/5 Hip internal rotators: 5/5 Knee extensors: 4 (vastus medialis: 4-) Knee flexors: 5/5 Ankle dorsiflexion: 5/5 Ankle plantarflexion: 5/5	51.9% left/48.1% right
Phase 2	Weeks 3–6	Knee flexion: 110°, Knee extension: 0°	1 (walking), 0 (at rest)	Hip flexors: 5/5 Hip extensors: 5/5 Hip abductors: 5/5 Hip adductors: 5/5 Hip external rotators: 5/5 Hip internal rotators: 5/5 Knee extensors: 5/5 (vastus medialis: 5) Knee flexors: 5/5 Ankle dorsiflexion: 5/5 Ankle plantarflexion: 5/5	Hip flexors: 5/5 Hip extensors: 5/5 Hip abductors: 4/5 Hip adductors: 5/5 Hip external rotators: 5/5 Hip internal rotators: 5/5 Knee extensors: 5- (vastus medialis: 4) Knee flexors: 5/5 Ankle dorsiflexion: 5/5 Ankle plantarflexion: 5/5	51.7% left/48.3% right
Phase 3	Weeks 7–12	Knee flexion: 120°, Knee extension: 0°	0 (walking), 0 (at rest)	Hip flexors: 5/5 Hip extensors: 5/5 Hip abductors: 5/5 Hip adductors: 5/5 Hip external rotators: 5/5 Hip internal rotators: 5/5 Knee extensors: 5/5 (vastus medialis: 5) Knee flexors: 5/5 Ankle dorsiflexion: 5/5 Ankle plantarflexion: 5/5	Hip flexors: 5/5 Hip extensors: 5/5 Hip abductors: 5/5 Hip adductors: 5/5 Hip external rotators: 5/5 Hip internal rotators: 5/5 Knee extensors: (vastus medialis: 5) Knee flexors: 5/5 Ankle dorsiflexion: 5/5 Ankle plantarflexion: 5/5	51.4% left/48.6% right

## Data Availability

All data are included in this study.

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
