# Peer review of "Accelerating Recovery: A Case Report on Telerehabilitation for a Triathlete’s Post-Meniscus Surgery Comeback"

_healthcare, 2025, doi:10.3390/healthcare13040406_

Round 1
Reviewer 1 Report
Comments and Suggestions for Authors
The topic of your article on “Accelerating Recovery: A Case Report on Tele-Rehabilitation for a Triathlete’s Post-Meniscus Surgery Comeback” is interesting and has the potential to bring insight into better adaptations of tele-rehabilitation (TR) in the everyday clinical practice with better outcomes and quicker recovery rate in athletes. Nevertheless, there are some aspects that should be addressed:
1. Please remain to either telerehabilitation or tele-rehabilitation: titles versus abstract: “ his case report explores the application of an innovative telerehabilitation…..”
The same applies to postoperative and post-operative : “…postoperative rehabilitation protocol after meniscal surgery. However, the post-operative…”
2. Please rewrite it accurately since you only refer to a male patient: “At the beginning of the third week and continuing through the fourth, the patient incorporated swimming into their routine”.
The same applies to “ This feature not only improved adherence to the rehabilitation protocol but also empowered the patient by providing objective feedback on their performance and recovery (18, 19). “
3. Please add more information on the surgical protocol, it is also important since this patient needed multidisciplinary approach.
4. Please make a table to add the outcomes (ROM, pain intensity, muscle strength etc) measured at different time points to be able to support the evolution of these parameters and make it easier to the readers to see these effects.
5. Regarding the “Telerehabilitation Program Details” I would split in what you aimed exactly to do as TR program part and the on-site rehabilitation program, you mixed them. Where all the exercises performed on the Khymeia system or also without it? I would like to see it more clearly explained, and if the exercises were mixed, on Khymeia and without, please state so.
6. When you make reference to Garmin devices stating that “This case illustrates the successful application of a telerehabilitation program enhanced by the integration of Garmin device data to achieve rapid and effective recovery in a triathlete following meniscal surgery” how were you able to assess these devices` effects for this patient? What outcomes did you take into consideration and how did you evaluate them? Please pay attention to any assumption you make, you need evidence to support it and you should change it in the manuscript accordingly.
7. Regarding the “Early Functional Gains: The patient achieved similar outcomes in terms of ROM, pain reduction, and functional strength when compared to traditional protocols.” How did you assess the patient`s response in comparison to a traditional protocol? Was the patient also included in another type of traditional/conventional rehabilitation program? Please state this in an accurate manner in the manuscript.
8. Also, since you state that the Garmin devices played such an important role in timely recovery, add more information about these devices, accessibility, and reliability to give a comprehensive understanding to the readers.
9. I suggest adding at least one figure showing the patient during his sessions on Khymeia system to add more relevance to your case report.
Good luck!
Author Response
REV 1
Comments and Suggestions for Authors
The topic of your article on “Accelerating Recovery: A Case Report on Tele-Rehabilitation for a Triathlete’s Post-Meniscus Surgery Comeback” is interesting and has the potential to bring insight into better adaptations of tele-rehabilitation (TR) in the everyday clinical practice with better outcomes and quicker recovery rate in athletes. Nevertheless, there are some aspects that should be addressed:
- Please remain to either telerehabilitation or tele-rehabilitation: titles versus abstract: “ his case report explores the application of an innovative telerehabilitation…..”
The same applies to postoperative and post-operative : “…postoperative rehabilitation protocol after meniscal surgery. However, the post-operative…”
Thank you for pointing this out. We have chosen to use the form without the hyphen ("telerehabilitation" and "postoperative") and ensured this consistency throughout the manuscript.
- Please rewrite it accurately since you only refer to a male patient: “At the beginning of the third week and continuing through the fourth, the patient incorporated swimming into their routine”.
The same applies to “ This feature not only improved adherence to the rehabilitation protocol but also empowered the patient by providing objective feedback on their performance and recovery (18, 19). “
Thank you for highlighting this point. We have revised the text to reflect the male gender of the patient, ensuring accuracy. The sentences have been updated as follows:
At the beginning of the third week and continuing through the fourth, the male patient incorporated swimming into his routine.
This feature not only improved adherence to the rehabilitation protocol but also empowered the patient by providing objective feedback on his performance and recovery.
- Please add more information on the surgical protocol, it is also important since this patient needed multidisciplinary approach.
Thank you for your comment, we have included the following information in the text. An informed consent was obtained for an arthroscopic procedure. The patient was operated in supine decubitus position with a pneumatic tourniquet at the root of the thigh (pressure at 250-300 mmHg) to optimize intra-articular visualization. Disinfection was obtained with povidone iodine and sterile drapes were used. Standard anteromedial and anterolateral arthroscopic portals were perfomed and an injury of the posterior horn of the medial meniscus was noted. Arthroscopy revealed also that there were superficial grade I International Cartilage Repair Society (ICRS) (9) erosions in the central portions of the medial tibial and femoral articular surfaces. No injuries of the anterior and posterior cruciate ligaments were reported. The tear of the medial meniscus appeared irreparabale and it was in the inner avascular two-thirds of the meniscus; thus, a selective partial meniscectomy was performed using basket forceps. A probe was used to evaluate mobility and instability of the injured tissue; the meniscal margins were with a motorized shaver to avoid unstable flaps and guarantee a regular edge.
Weight bearing was allowed as tolerated with use of crutches until normal gate was achieved two weeks postoperatively (10, 11). Cryotherapy was prescribed for 15 minutes four times a day and deep vein thrombosis prophylaxis was performed using low-molecular-weight heparin (12, 13). The patient recovered uneventfully.
- Please make a table to add the outcomes (ROM, pain intensity, muscle strength etc) measured at different time points to be able to support the evolution of these parameters and make it easier to the readers to see these effects.
Thank you for your feedback. As per your suggestion, we have created a table to present the outcomes (e.g., ROM, pain intensity, muscle strength, grond contact time) measured at different time points. This addition allows for a clearer visualization of the evolution of these parameters, making it easier for readers to interpret the effects. The table has been incorporated into the manuscript accordingly.
Phase |
Time Period |
ROM |
Pain Intensity (VAS) |
Muscle Strength Left (MRC Scale) |
Muscle Strength Right (MRC Scale) |
Biomechanical Data (Ground Contact Time Distribution) |
Preoperative |
Before surgery |
Knee flexion: 110°, Knee extension: -10° |
3 (walking), 0 (at rest) |
Hip flexors: 5/5 |
Hip flexors: 5/5 |
51.5% left / 48.5% right |
Phase 1 |
Weeks 1–2 |
Knee flexion: 90°, Knee extension: -5° |
2 (walking), 0 (at rest) |
Hip flexors: 5/5 |
Hip flexors: 5/5 |
51.9% left / 48.1% right |
Phase 2 |
Weeks 3–6 |
Knee flexion: 110°, Knee extension: 0° |
1 (walking), 0 (at rest) |
Hip flexors: 5/5 |
Hip flexors: 5/5 |
51.7% left / 48.3% right |
Phase 3 |
Weeks 7–12 |
Knee flexion: 120°, Knee extension: 0° |
0 (walking), 0 (at rest) |
Hip flexors: 5/5 |
Hip flexors: 5/5 |
51.4% left / 48.6% right |
- Regarding the “Telerehabilitation Program Details” I would split in what you aimed exactly to do as TR program part and the on-site rehabilitation program, you mixed them. Where all the exercises performed on the Khymeia system or also without it? I would like to see it more clearly explained, and if the exercises were mixed, on Khymeia and without, please state so.
We have added the requested information to the text, clearly distinguishing between the telerehabilitation program and the on-site rehabilitation program. Additionally, we have specified which exercises were performed using the Khymea system and which were conducted without it.
- When you make reference to Garmin devices stating that “This case illustrates the successful application of a telerehabilitation program enhanced by the integration of Garmin device data to achieve rapid and effective recovery in a triathlete following meniscal surgery” how were you able to assess these devices` effects for this patient? What outcomes did you take into consideration and how did you evaluate them? Please pay attention to any assumption you make, you need evidence to support it and you should change it in the manuscript accordingly.
Thank you for your insightful comment. We acknowledge the need to clarify the role of wearable devices in our assessment and ensure that our conclusions are evidence-based. In our case, the wearable device was used primarily to track specific objective metrics related to the patient’s rehabilitation progress, including:
- Heart rate monitoring: To assess cardiovascular response during swimming, cycling, and running sessions.
- Training duration and cadence: To evaluate workload progression in cycling and running.
- Running biomechanics: Specifically, ground contact time distribution, which provided insights into potential asymmetries between the operated and non-operated limb.
The data collected from the wearable device was used as an adjunct to standard physiotherapy assessments rather than as a primary outcome measure. These parameters were integrated into the rehabilitation monitoring process to adjust exercise intensity and detect potential imbalances that could influence recovery.
To address your concern, we have revised the manuscript to clarify that:
- The wearable device was used for objective tracking of physiological and biomechanical parameters rather than as a direct measure of recovery success.
- The interpretation of its data was integrated with clinical assessments (ROM, pain, strength tests, and functional performance) conducted by the physiotherapist.
- Any claims regarding its role have been modified to ensure that our conclusions are fully evidence-based and not overstated.
- Following the suggestion of another reviewer, we have generalized the term to "wearable device" instead of referring to a specific brand, as the collected data does not strictly depend on the manufacturer.
We appreciate your valuable feedback, which has helped us refine our manuscript to enhance clarity and scientific rigor.
- Regarding the “Early Functional Gains:The patient achieved similar outcomes in terms of ROM, pain reduction, and functional strength when compared to traditional protocols.” How did you assess the patient`s response in comparison to a traditional protocol? Was the patient also included in another type of traditional/conventional rehabilitation program? Please state this in an accurate manner in the manuscript.
Thank you for your insightful comment. In this case report, the patient did not participate in a traditional rehabilitation program alongside the telerehabilitation protocol. Instead, the patient's progress was evaluated using measures like range of motion (ROM), pain levels (VAS scale), and muscle strength (MRC scale), which were compared to reported outcomes from conventional rehabilitation protocols in the literature. These comparisons provided a benchmark to assess the telerehabilitation program's effectiveness in achieving similar functional gains. We will revise the manuscript to clarify this methodology and explicitly state that the comparison was based on data from existing studies rather than a direct side-by-side evaluation.
- Also, since you state that the Garmin devices played such an important role in timely recovery, add more information about these devices, accessibility, and reliability to give a comprehensive understanding to the readers.
Thank you for your comment. Garmin devices, and more generally, wearable devices, are widely accessible and known for their reliability in tracking physical activity, including heart rate, swimming stroke efficiency, cycling cadence, and running biomechanics. It is important to note that our patient already owned the device prior to the rehabilitation program, and it was not specifically acquired for rehabilitation purposes.
- I suggest adding at least one figure showing the patient during his sessions on Khymeia system to add more relevance to your case report.
Thank you for your suggestion. We have included a figure in the text showing the patient during his sessions on the Khymeia system to enhance the relevance of the case report.

Reviewer 2 Report
Comments and Suggestions for Authors
The title is very interesting. Thank you for posibility to read it. The primary objective is to demonstrate the feasibility and effectiveness of an early initiated telerehabilitation program tailored to support a rapid return to sports. There are too many variables that can affect the final result, even diet. Is it possible to determine what influenced this knee and how? The causes of the injury were described in great detail. This probably has no significance or impact on the final effect of telerehabilitation and is not needed in the manuscript. Maybe the authors see some connections here, but I don't. The methods of collecting data by the devices used were not discussed, but its brand was clearly emphasized. Is this necessary? Would the use of a different measurement or control tool give a different result of telerehabilitation? The authors try to compare and even show the superiority of one form of rehabilitation over another, but they do not have a control group and cannot compare anything. I think that the topic and the entire script are worth the reader's interest, but descriptions of the accident that do not affect the result should be limited. The measurement methods need to be described in more detail. Conclusions should be limited to what has actually been examined. In its current version, it is an interesting story, but it needs to be made credible.
Author Response
REV2
The title is very interesting. Thank you for posibility to read it. The primary objective is to demonstrate the feasibility and effectiveness of an early initiated telerehabilitation program tailored to support a rapid return to sports. There are too many variables that can affect the final result, even diet. Is it possible to determine what influenced this knee and how? The causes of the injury were described in great detail. This probably has no significance or impact on the final effect of telerehabilitation and is not needed in the manuscript. Maybe the authors see some connections here, but I don't. The methods of collecting data by the devices used were not discussed, but its brand was clearly emphasized. Is this necessary? Would the use of a different measurement or control tool give a different result of telerehabilitation? The authors try to compare and even show the superiority of one form of rehabilitation over another, but they do not have a control group and cannot compare anything. I think that the topic and the entire script are worth the reader's interest, but descriptions of the accident that do not affect the result should be limited. The measurement methods need to be described in more detail. Conclusions should be limited to what has actually been examined. In its current version, it is an interesting story, but it needs to be made credible.
Influence of Variables on Outcomes: We agree that various factors, such as diet, can influence rehabilitation outcomes. While the patient's nutritional regimen was documented as part of the telerehabilitation program, we acknowledge that isolating its impact was not feasible within the scope of this case report. We have clarified this limitation in the revised manuscript and emphasized the multifactorial nature of recovery.
We also acknowledge that multiple factors, including the patient’s diet, exercise adherence, and preoperative physical condition, can influence rehabilitation outcomes. While the patient’s nutritional regimen was documented as part of the program, isolating its specific contribution to recovery was not feasible within the scope of this case. The multifactorial nature of recovery underscores the importance of considering these elements holistically, so this is a possible limit of our study.
Detailed Description of Injury Causes: While the patient's injury mechanism was described to provide context, we agree that its level of detail may not directly contribute to the telerehabilitation outcomes. We have streamlined this section to focus on clinically relevant aspects: In May 2024, the patient sustained a medial meniscus tear during an off-road running session, resulting in swelling and posteromedial knee pain that worsened with activity.
Data Collection Methods and Device Emphasis: We appreciate your point regarding the lack of detail on data collection methods. In response, we have expanded the description of how data from the devices (e.g., motion detection and activity monitoring tools) were collected, processed, and integrated into the rehabilitation protocol. Regarding the brand mention, we recognize the potential for bias and have reframed this to focus on device functionality rather than brand names, while maintaining necessary clarity for reproducibility.
Impact of Different Measurement Tools: We agree that discussing the potential impact of alternative tools would strengthen the manuscript. We have added a brief discussion on how varying device functionalities might influence rehabilitation monitoring and outcomes.
Comparison Without a Control Group: We acknowledge that the absence of a control group limits the strength of our conclusions. The comparison with traditional rehabilitation protocols in the discussion was meant to contextualize the outcomes rather than claim superiority. We have rephrased these sections to avoid overinterpretation and to emphasize the exploratory nature of this case report.
Scope of Conclusions: We concur that conclusions should be limited to what was directly examined. We have revised the conclusion to reflect only the findings derived from this single case, emphasizing the need for larger studies to validate these observations.

Reviewer 3 Report
Comments and Suggestions for Authors
Dear authors
Thank you for sending this manuscript to Healthcare.
You paper seems to be interesting for readers but there are many problems as below.
Lines: 64-65: “Given the limited existing literature, a comprehensive description of the recommended rehabilitation following this surgical procedure is essential.”
-Is this claim by the respected authors correct? Basically, such a claim is not acceptable after conducting a case study with a volunteer.
Lines: 69-70: The authors wrote::“The primary objective is to demonstrate the feasibility and effectiveness of an early initiated telerehabilitation program tailored to support a rapid return to sports for a 49- year-old triathlete following meniscus surgery.” Then In Line 76 again you wrote “The patient is a healthy 49-year-old male triathlete”
-So, which one is correct?? A patient with meniscus injury, or a healthy subject???
Lines 98-100: “”His regimen typically included multiple weekly sessions spanning the three disciplines of triathlon: swimming (2-3 times a week), cycling (3 times a week), and running (3-4 times a week), with a focus on enhancing cardiovascu lar fitness and muscular endurance.”””
-all these regimen needed to have reference(s). Here, it is necessary to use an exercise protocol or regimen endorsed in past journals and articles.
Line 105: “Khymeia System (Padova, Italy),”
- Is this system is valid and reliable????
Lines 130-131 “”All exercises were performed while respecting the weight-bearing limitations and maintaining knee flexion within a 90-degree range. The exercises were performe..””
- Here, again it is necessary to use an exercise protocol or regimen endorsed in past journals and articles.
Lines 140-142: it is needed to use refrence(s)
The study discussion is quite brief and does not seem to be able to answer the research conducted.
Best
Comments on the Quality of English LanguageIt is better to edit.
Author Response
REV 3
Dear authors
Thank you for sending this manuscript to Healthcare.
You paper seems to be interesting for readers but there are many problems as below.
Lines: 64-65: “Given the limited existing literature, a comprehensive description of the recommended rehabilitation following this surgical procedure is essential.”
-Is this claim by the respected authors correct? Basically, such a claim is not acceptable after conducting a case study with a volunteer.
Thank you for your comment. We have modified in to the text.
Lines: 69-70: The authors wrote::“The primary objective is to demonstrate the feasibility and effectiveness of an early initiated telerehabilitation program tailored to support a rapid return to sports for a 49- year-old triathlete following meniscus surgery.” Then In Line 76 again you wrote “The patient is a healthy 49-year-old male triathlete”
-So, which one is correct?? A patient with meniscus injury, or a healthy subject???
Thank you for your comment. We originally described the patient as 'healthy' to indicate his overall good physical condition. However, following the reviewer's feedback, we have removed this term to avoid confusion
Lines 98-100: “”His regimen typically included multiple weekly sessions spanning the three disciplines of triathlon: swimming (2-3 times a week), cycling (3 times a week), and running (3-4 times a week), with a focus on enhancing cardiovascu lar fitness and muscular endurance.”””
-all these regimen needed to have reference(s). Here, it is necessary to use an exercise protocol or regimen endorsed in past journals and articles.
Thank you for your comment, but the described training regimen reflects the athlete’s habitual practice and personal preferences rather than a standardized protocol. It aligns with the typical training loads observed in competitive triathletes but was individually structured by the patient based on his experience and competition goals.
Line 105: “Khymeia System (Padova, Italy),”
- Is this system is valid and reliable????
Khymeia System (Padova, Italy), a digital platform designed for remote rehabilitation, has been validated in previous studies for its reliability in motor rehabilitation and its effectiveness in telemonitoring-based physiotherapy
Lines 130-131 “”All exercises were performed while respecting the weight-bearing limitations and maintaining knee flexion within a 90-degree range. The exercises were performe..””
- Here, again it is necessary to use an exercise protocol or regimen endorsed in past journals and articles.
Thank you for your comment. We would like to clarify that the weight-bearing limitations and knee flexion range were not determined by the physiotherapist but were strict postoperative recommendations from the surgeon. Adhering to these guidelines is standard practice to ensure the safety of the patient and to promote optimal recovery
Lines 140-142: it is needed to use refrence(s)
Thank you for your comment. The statement in question reports clinical observations directly assessed in the patient, based on standard measurements used in post-meniscectomy rehabilitation. The values for knee extension, flexion, and pain on the VAS scale are objective data collected during patient monitoring and fall within the ranges typically described in the literature for the early postoperative rehabilitation phase, however an "Outcome" section has been included, featuring a summary table detailing the results achieved across the different phases of rehabilitation and relevant references supporting the selected outcome measures (such as VAS, ROM, MRC scale, ground contact time) have been incorporated into this section to enhance the study's methodological rigor.
The study discussion is quite brief and does not seem to be able to answer the research conducted.
Thank you for your comment, we have enriched the discussion section.

Reviewer 4 Report
Comments and Suggestions for Authors
The article is only a case report, but a larger scale controlled study is needed to evaluate the effectiveness of telerehabilitation protocols. The lack of a randomized controlled trial limits the generalizability of the results.
The assessment of rehabilitation progress is based solely on subjective reports and individual measurements. The use of more objective metrics (e.g., biomechanical analysis, EMG data, etc.) could have provided more reliable results.
Statistical analysis of the data obtained from the study is lacking. For example, no analysis was conducted to evaluate the significance levels of improvements in specific measurements.
A literature review could have provided a more comprehensive context for the effectiveness of telerehabilitation in athletes. Some of the references in the study may be outdated and do not benefit from current developments.
Although the effects of nutritional strategies are described in detail, there is no method to analyze the effects of this protocol in isolation. The effects of nutritional and exercise protocols should have been measured separately.
The visuals used (e.g., demonstrations of exercises) could have been clearer and more detailed. This would have provided readers with a better understanding of the rehabilitation process.
Author Response
REV 4
The article is only a case report, but a larger scale controlled study is needed to evaluate the effectiveness of telerehabilitation protocols. The lack of a randomized controlled trial limits the generalizability of the results.
Thank you for your comment. As noted in the 'Conclusion and Limitations' section, we acknowledge that this study is a case report and that further large-scale, controlled trials are needed to assess the effectiveness of telerehabilitation protocols. We explicitly state the limitations of our work, including the lack of a randomized controlled trial, which affects the generalizability of the findings. However, we believe that this case contributes valuable preliminary insights that can help inform future research in this field.
The assessment of rehabilitation progress is based solely on subjective reports and individual measurements. The use of more objective metrics (e.g., biomechanical analysis, EMG data, etc.) could have provided more reliable results.
Thank you for your valuable comment. While we acknowledge the importance of objective metrics such as biomechanical analysis or EMG data, it is worth noting that subjective measures, including pain scales, patient-reported outcomes, and clinical assessments, are widely used and validated in the orthopedic rehabilitation literature. In our study, we complemented these subjective assessments with objective data from wearable devices, which allowed real-time monitoring of functional parameters such as ground contact time distribution. Nevertheless, we recognize that future studies could benefit from incorporating additional biomechanical analyses to further refine telerehabilitation protocols
Statistical analysis of the data obtained from the study is lacking. For example, no analysis was conducted to evaluate the significance levels of improvements in specific measurements.
We appreciate the reviewer’s comment. As this is a single-case report, statistical analysis was not applicable, given the nature of the study design. The primary aim was to document the rehabilitation process and outcomes in detail, providing insights for future research. Nonetheless, we recognize the importance of statistical validation in broader studies and have already highlighted in the limitations section the need for larger controlled trials to further evaluate the effectiveness of this approach.
A literature review could have provided a more comprehensive context for the effectiveness of telerehabilitation in athletes. Some of the references in the study may be outdated and do not benefit from current developments.
Thank you for your suggestion. While this case report primarily focuses on the feasibility and outcomes of a telerehabilitation program for a triathlete post-meniscus surgery, we acknowledge the value of a broader literature review. In the introduction and discussion, we referenced key studies that contextualize our findings within the existing body of research on telerehabilitation and post-meniscectomy rehabilitation. However, we appreciate your feedback and will consider expanding the review in future work to include the most recent advancements in the field. Regarding the references, we have ensured that they are relevant to the study’s objectives, but we will carefully reassess them to integrate more recent developments where appropriate.
Although the effects of nutritional strategies are described in detail, there is no method to analyze the effects of this protocol in isolation. The effects of nutritional and exercise protocols should have been measured separately.
Thank you for your insightful comment. In our study, we aimed to adopt a holistic approach to rehabilitation, combining exercise and nutritional strategies, as these factors are inherently interrelated in optimizing recovery. We acknowledge that isolating the effects of each component would require a different study design, such as a randomized controlled trial with separate intervention groups. However, the primary objective of this case report was to describe the feasibility and potential benefits of an integrated telerehabilitation protocol, reflecting real-world rehabilitation practices. Future studies could explore the individual impact of nutritional strategies versus exercise interventions in a more controlled setting.
The visuals used (e.g., demonstrations of exercises) could have been clearer and more detailed. This would have provided readers with a better understanding of the rehabilitation process.
Thank you for your feedback. We chose to use the images from the Khymeia database because they represent exercises available in the database, this not only ensures consistency with the protocol described in the case report but also provides healthcare professionals with concrete references for designing rehabilitation programs based on tools that are already available and used in clinical practice. However, we take note of your suggestion and will consider possible improvements to make the visual content even clearer and more detailed in future publications.

Round 2
Reviewer 1 Report
Comments and Suggestions for Authors
Dear authors,
I appreciate your commitment to meeting the recommendations and thank you for your clarifications, you enhanced the relevance of the case report. I appreciate you included the modifications suggested in the manuscript, and I am confident that your manuscript will easily attain its aim for the readers.
Good luck!
Author Response
Thank you very much for your positive feedback and encouraging words. We truly appreciate your valuable suggestions and insightful comments, which have significantly improved the quality and relevance of our case report.
We are grateful for your time and effort in reviewing our manuscript and for your confidence in its potential impact on the readers.
Warm regards,
Marina Garofano

Reviewer 2 Report
Comments and Suggestions for Authors Thank you for all your replies. I see that telerehabilitation may be important, but the changes are so small that it should not surprise anyone. I see that many scientists worked on this project and I even regret that only one patient was followed. It is a pity that measurements that would be more measurable and reliable were not performed.Author Response
Thank you for your thoughtful feedback and for taking the time to review our manuscript. We appreciate your constructive comments, which help us reflect critically on our work.
We acknowledge your concerns regarding the small changes observed through telerehabilitation. While the case report focuses on a single patient, our aim was to highlight the feasibility and practical application of telerehabilitation, particularly in high-performance athletes where even minor improvements can be clinically significant.
Regarding the inclusion of only one patient, we understand your perspective. However, as a case report, the study was designed to offer an in-depth analysis of an individual experience rather than generalized outcomes. We believe this detailed observation provides valuable insights into personalized rehabilitation strategies and sets the groundwork for future studies with larger cohorts.
We also recognize the limitations concerning measurable and more objective data. While we employed standardized tools like the VAS for pain, MRC scale for strength, and ground contact time distribution, we agree that incorporating additional objective measures (e.g., isokinetic dynamometry or motion analysis) could have strengthened the findings. We have highlighted this as a limitation in the manuscript and suggested directions for future research.
Thank you again for your constructive critique, which will undoubtedly help improve the manuscript's clarity and impact.
Best regards,
Marina Garofano

Reviewer 3 Report
Comments and Suggestions for Authors
Dear authors
Thank you for your corrections.
The only question that your answer didn't satisfy me was this one:
""-Is this claim by the respected authors correct? Basically, such a claim is not acceptable after conducting a case study with a volunteer.
Your Answer! Thank you for your comment. We have modified in to the text."
Anyway, if the Associate Editor OR the Editor-in-Chief accept your answer, I think it would be fair to publish this article, but overall, this answer was not convincing to me.
Best
Author Response
Thank you for your valuable feedback and for sharing your observations regarding the specific claim in our manuscript. We sincerely appreciate your critical comments, which are essential for improving the clarity and robustness of our work.
We apologize if our previous response appeared superficial and did not fully meet your expectations. We acknowledge the importance of the issue you raised and would like to clarify our position more thoroughly. The claim in question has been carefully revised to more accurately reflect the limitations inherent in a single case study. We have explicitly highlighted that our conclusions are based on observations from this specific case and should not be generalized without further validation through studies involving larger cohorts.
Our goal was to emphasize the potential of telerehabilitation in the recovery process of high-demand athletes following meniscus surgery, based on the positive outcomes observed in this case. However, we have now reformulated the text to ensure that the exploratory nature of this case report is clearly conveyed, avoiding any overgeneralization.
Thank you again for your time and for your significant contribution to improving our manuscript.
Kind regards,
Marina Garofano

Reviewer 4 Report
Comments and Suggestions for Authors
With the revisions, the article has been sufficiently improved and is ready for publication.
Author Response
Thank you for your positive feedback and for taking the time to review our manuscript. We greatly appreciate your constructive comments and insightful suggestions, which have contributed to improving the quality of our work.
We are pleased to hear that you find the revisions satisfactory and that the article is now ready for publication. Your support and thoughtful evaluation have been invaluable throughout this process.
Warm regards,
Marina Garofano
